# CLIMATELLM: EFFICIENT WEATHER FORECASTING VIA FREQUENCY-AWARE LARGE LANGUAGE MODELS

## ABSTRACT

Recent progress in deep learning has advanced global weather forecasting, with larger and higher-resolution models steadily improving skill. In parallel, spectral methods provide an efficient basis for global dynamics. Yet most spectral approaches treat the complex spectrum as generic features, conflating the distinct physics encoded in amplitude (energy evolution) and phase (spatial propagation). **We propose ClimateLLM, a physics-aligned, frequency-domain forecasting framework powered by SAED-Former**. At its core, **SAED-Former** explicitly separates these two processes via a *dual-state representation*, computes interactions through a *phase-centric propagation kernel*, and injects wave-number–aware priors using *scale-conditional projection*. This physics-aligned design yields compact, robust frequency-domain representations. On standard reanalysis benchmarks, ClimateLLM matches or exceeds state-of-the-art accuracy across short- and medium-range horizons while training on a single GPU within hours. Moreover, the model supports *cross-variable transference*: networks trained on data-rich variables produce robust zero-shot forecasts for data-scarce variables. By elevating spectral structure to first-class status, ClimateLLM improves forecast quality, efficiency, and generalization.

## 1 INTRODUCTION

Accurate and timely weather forecasting underpins climate adaptation, disaster mitigation, and economic planning (Bi et al., 2023; Pathak et al., 2022). Recent advances in deep learning have delivered strong gains, often by scaling models and input resolution (Chen et al., 2023a; Lam et al., 2023). While effective, this scale-first trajectory raises computational cost and data demand, and it does not directly address three pressing needs for weather foundation models: **computational efficiency**, **long-horizon temporal reasoning**, and **generalization in low-data regimes**, such as **zero-shot cross-variable forecasting**.

Spectral methods offer a principled path toward efficiency. Operating in the frequency domain provides global basis functions that compactly capture planetary-scale teleconnections central to large-scale atmospheric evolution (Pathak et al., 2022; Li et al., 2020). However, most spectral deep models exhibit a **structural blindness**: they treat the complex-valued spectrum as an unstructured feature vector. This collapses two physically distinct processes—**energy evolution (amplitude)** and **spatial propagation (phase)**—into a single channel, forcing networks to learn disentanglement implicitly. The consequences include brittle generalization, inflated data and compute requirements, and limited transfer across variables.

We advance a shift from structurally-blind regression to **structurally-aware dynamics modeling** in the spectral domain (Yin et al., 2021; Raissi et al., 2019). The principle is to elevate spectral structure to first-class status by explicitly decoupling amplitude and phase throughout the model. Concretely, we build **ClimateLLM** on a new backbone, the **Scale-Aware Entangled Dynamics Transformer (SAED-Former)**, that operationalizes three design elements. First, a **dual-state representation** maintains parallel states for energy evolution (amplitude) and spatial propagation (phase) (Tran et al., 2021). Second, a **phase-centric propagation kernel** governs interactions based solely on propagation dynamics, allowing the model to aggregate information along physically meaningful pathways. Third, a **scale-conditional projection** injects wave-number–aware priors so the network can specialize dynamics across atmospheric scales (Chattopadhyay et al., 2020). Together, these

components instantiate a compact, physics-aligned representation that the network can learn efficiently.

This design directly targets the three axes above. By separating energy and propagation, the backbone reduces the burden of implicit disentanglement, improving **computational efficiency** and enabling **single-GPU, hours-level training**. By encoding propagation pathways at the operator level, it naturally accommodates **long-horizon inputs**. By preserving a physically meaningful factorization, it supports **cross-variable transference**: a model trained on data-rich variables (e.g., temperature) can produce robust **zero-shot** forecasts for data-scarce variables (e.g., soil moisture), expanding applicability where labeled data are limited.

**Contributions.**

- We diagnose the **structural blindness** of current spectral forecasting—the conflation of amplitude (energy evolution) and phase (spatial propagation)—and propose a **structurally-aware dynamics** paradigm that makes this duality explicit.

- We introduce **SAED-Former**, which instantiates this paradigm via (i) a **dual-state representation**, (ii) a **phase-centric propagation kernel**, and (iii) a **scale-conditional projection** that encodes scale-dependent physics.

- We demonstrate that **ClimateLLM** attains state-of-the-art accuracy with substantially lower compute, supports **zero-shot** cross-variable forecasting, and scales to **longer horizons**, indicating a practical route to efficient, generalizable weather foundation models.

## 2 RELATED WORK

### 2.1 DEEP LEARNING MODELS FOR WEATHER FORECASTING

Deep learning models have demonstrated significant advantages over traditional numerical weather forecasting methods Leinonen et al. (2023); Li et al. (2024); Salman et al. (2015); Hewage et al. (2021). FourCastNet Pathak et al. (2022) outperforms traditional systems in predicting small-scale variables and extreme weather events while using a fraction of the computational resources. Graph-Cast Lam et al. (2022) delivers highly accurate 10-day global forecasts in under a minute, excelling at severe weather prediction. GenCast Price et al. (2023) provides more accurate and efficient probabilistic forecasts than ECMWF's ensemble approach Molteni et al. (1996). FuXi Chen et al. (2023b) offers 15-day global forecasts matching ECMWF's ensemble mean while extending the skillful forecast period. Other deep learning time series models have also shown promise in temporal forecasting tasks Zhou et al. (2022); Zhang & Yan (2023); Eldele et al. (2024); Yi et al. (2024).

Pathak et al. Pathak et al. (2022) apply Adaptive Fourier Neural Operators to learn weather variable evolution across spatial and temporal domains, capturing both large-scale trends and fine-grained structures. Sun Sun et al. (2023) employs FNOs as surrogate models to predict flood extents and water depths at high resolution. FNOs efficiently simulate fluid dynamics through global convolution, making them ideal for long-term trend modeling and data-driven forecasting of meteorological phenomena.

### 2.2 LARGE LANGUAGE MODELS FOR TIME-SERIES PREDICTION

Large language models (LLMs) have proven effective for time series forecasting Chang et al. (2023); Sun et al. (2024). TIME-LLM Jin et al. (2023) aligns time series with language modalities, outperforming specialized forecasting models. The Frozen Pretrained Transformer Zhou et al. (2023) achieves state-of-the-art results across various time series tasks using pre-trained models. CALF Liu et al. (2024) reduces distribution discrepancies between textual and temporal data, improving LLM performance in forecasting tasks. TEMPOCao et al. (2023) introduces a prompt-based generative transformer that decomposes time series into trend, seasonal and residual components, achieving state-of-the-art results in zero-shot forecasting tasks through effective knowledge transfer from pretrained language models to temporal data.

LLMs show particular promise for weather forecasting applications Wang & Karimi (2024); Wang et al. (2024); Li et al. (2024). Li et al. Li et al. (2024) introduce CLLMate, a multimodal LLM using meteorological raster data and textual event data for climate forecasting.

# 3 SCALE-AWARE ENTANGLED DYNAMICS FOUNDATION MODEL

Current spectral weather models typically treat atmospheric evolution as a black-box mapping from historical to future states, overlooking the distinct physical processes governing the dynamics. We propose that robust and interpretable forecasting requires explicitly decoupling atmospheric evolution into two fundamental processes: **Energy Evolution** (governing intensification/dissipation) and **Spatial Propagation** (governing movement/translation). These processes exhibit **scale-dependent** behavior—planetary waves and convective systems evolve distinctly.

We introduce **ClimateLLM**, built on the **Scale-Aware Entangled Dynamics Transformer (SAED-Former)**, a physically-principled architecture that separately models these entangled dynamics while maintaining scale awareness.

## 3.1 PRELIMINARIES: DYNAMICS DUALITY IN THE FREQUENCY DOMAIN

Our approach is founded upon representing the atmospheric state in the frequency domain. We select the two-dimensional Fast Fourier Transform (FFT) not merely for its computational efficiency, but for its unique ability to decompose a spatial field into a basis of global sinusoidal waves. This global perspective is crucial for capturing large-scale teleconnections, a feature that local basis functions like wavelets may obscure.

Given a spatial weather field $\mathbf{X}_t \in \mathbb{R}^{H \times W}$ at a discrete time step $t$, where $H$ and $W$ are the height and width of the grid, its frequency domain representation $\mathbf{F}_t \in \mathbb{C}^{H \times W}$ is obtained via FFT:

$$\mathbf{F}_t = \mathcal{F}(\mathbf{X}_t) \tag{1}$$

Each complex coefficient $F_{t,(k_x,k_y)} \in \mathbf{F}_t$ at wavenumber $(k_x, k_y)$ can be further decomposed into its polar components, revealing a fundamental duality of the system's dynamics:

$$F_{t,(k_x,k_y)} = A_{t,(k_x,k_y)} \cdot e^{iP_{t,(k_x,k_y)}} \tag{2}$$

where $A_{t,(k_x,k_y)} \in \mathbb{R}^+$ and $P_{t,(k_x,k_y)} \in [-\pi, \pi)$ represent the amplitude and phase, respectively. We establish a direct mapping from these components to physical processes:

- **Amplitude Spectrum ($\mathbf{A}_t$):** The magnitude of the spectral coefficients, where $A_{t,(k_x,k_y)} = |F_{t,(k_x,k_y)}|$, corresponds to the **energy** of a weather system at a specific spatial scale. The evolution of $\mathbf{A}_t$ thus represents the system's intensification or dissipation dynamics.

- **Phase Spectrum ($\mathbf{P}_t$):** The argument of the coefficients, where $P_{t,(k_x,k_y)} = \arg(F_{t,(k_x,k_y)})$, encodes the spatial position and alignment of the wave components. The evolution of $\mathbf{P}_t$ therefore represents the system's spatial **propagation**, including translation and rotation.

The core challenge, therefore, is to design a neural network that can comprehend this duality and learn to predict the distinct, yet entangled, temporal evolutions of $\mathbf{A}_t$ and $\mathbf{P}_t$.

## 3.2 OVERALL ARCHITECTURE

The SAED-Former is designed as a sequence-to-sequence model that operates autoregressively in the frequency domain. As illustrated in Figure 1, the model takes a sequence of past weather states as input and predicts the state for the subsequent time step. This process is orchestrated through three main stages: input representation, sequential processing via a stack of our novel SAED-Former blocks, and output reconstruction.

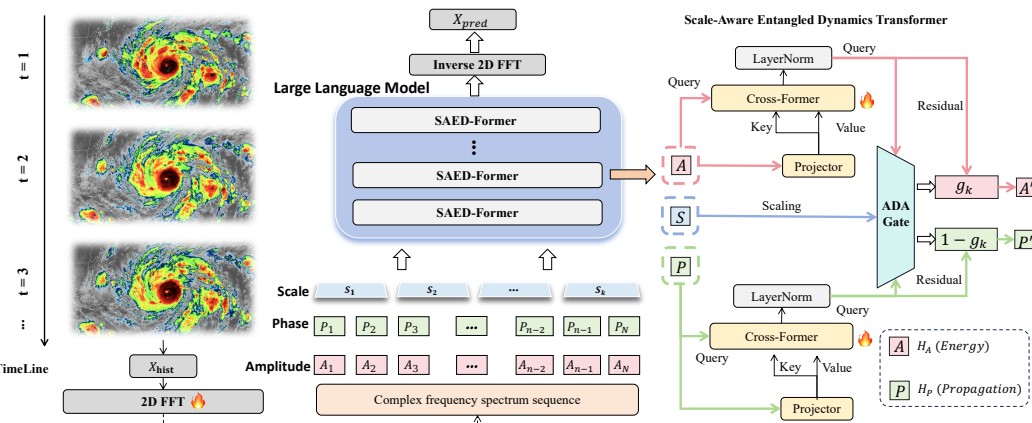

Figure 1: The overall architecture of the Scale-Aware Entangled Dynamics Transformer (SAED-Former). A sequence of historical weather maps is transformed into the frequency domain and augmented with scale embeddings. This representation is then processed by a stack of $L$ SAED-Former blocks, each performing an entangled update of the system's energy and propagation states. The final predicted spectrum is transformed back to the spatial domain.

**Input Representation.** Given past $\mathcal{T}$ observations $\{\mathbf{X}_{t-\mathcal{T}+1}, \ldots, \mathbf{X}_t\}$ with $\mathbf{X} \in \mathbb{R}^{H \times W}$, we apply FFT to obtain complex spectra $\{\mathbf{F}_{t-\mathcal{T}+1}, \ldots, \mathbf{F}_t\}$, $\mathbf{F} \in \mathbb{C}^{H \times W}$. Each spectrum is flattened into tokens indexed by wavenumber $(k_x, k_y)$. To encode scale, we partition the frequency plane into $K$ disjoint bands by wavenumber magnitude and define a band-mapping $\mathcal{B} : \mathbb{Z}^2 \to \{1, \ldots, K\}$. Each token retrieves a learnable scale vector

$$\mathbf{s}_{(k_x, k_y)} = \mathbf{E}_{\text{scale}}[\mathcal{B}(k_x, k_y)], \quad \mathbf{E}_{\text{scale}} \in \mathbb{R}^{K \times d_s}. \tag{3}$$

The scale vector is concatenated to token features derived from $F_{t, (k_x, k_y)}$.

**Core Processing via SAED-Former Blocks.** The $\mathcal{T}$ sequences of augmented tokens feed a decoder-only Transformer (GPT-2 style (Radford et al., 2019)) composed of $L$ identical **SAED-Former Blocks**. Each block takes dual-state sequences from the previous layer, applies a **phase-centric propagation kernel** (interactions governed by phase) and a **Scale-Aware Evolution Module (SAEM)** (scale-conditioned updates), then outputs the next-layer dual states. The top block produces the dual-state representation used to forecast step $t+1$.

**Dual-State Hidden Representation.** For each token $k$, we maintain two $d_{\text{model}}$-dimensional states:

$$\mathbf{h}_{A,k} \in \mathbb{R}^{d_{\text{model}}} \quad \textbf{Energy State}, \qquad \mathbf{h}_{P,k} \in \mathbb{R}^{d_{\text{model}}} \quad \textbf{Propagation State}.$$

They are initialized from $F_k = A_k e^{iP_k}$ and updated jointly across layers: the Energy State focuses on amplitude $A_k$ (energy evolution), while the Propagation State captures phase $P_k$ (spatial propagation).

### 3.2.1 Phase-Centric Attention as a Propagation-Kernel Operator

**Purpose.** We define a **Propagation-Kernel Operator** that drives interactions *exclusively* by the **spatial propagation (phase)** representation, and uses these interactions to update the dual-state sequence $\mathfrak{H} = (\mathbf{H}_A, \mathbf{H}_P)$, where $\mathbf{H}_A, \mathbf{H}_P \in \mathbb{R}^{N \times d_{\text{model}}}$.

**Propagation Similarity Kernel.** Let $\phi_q, \phi_k : \mathbb{R}^{d_{\text{model}}} \to \mathbb{R}^{d_k}$ be learned projections applied to propagation states. For tokens $i, j$, define

$$\kappa(\mathbf{h}_{P,i}, \mathbf{h}_{P,j}; \mathbf{H}_P) = \frac{\exp\left(\frac{\langle \phi_q(\mathbf{h}_{P,i}), \phi_k(\mathbf{h}_{P,j}) \rangle}{\sqrt{d_k}}\right)}{\sum_{l=1}^{N} \exp\left(\frac{\langle \phi_q(\mathbf{h}_{P,i}), \phi_k(\mathbf{h}_{P,l}) \rangle}{\sqrt{d_k}}\right)}, \tag{4}$$

and collect $\alpha_{ij} = \kappa(\mathbf{h}_{P,i}, \mathbf{h}_{P,j}; \mathbf{H}_P)$ into $\boldsymbol{\alpha} \in \mathbb{R}^{N \times N}$. Thus, the **field of influence $\boldsymbol{\alpha}$** is a pure function of **propagation (phase)**.

**Dual-State Update Operator.** Let $\psi_A, \psi_P : \mathbb{R}^{d_{\text{model}}} \to \mathbb{R}^{d_v}$ be value maps for **energy (amplitude)** and **propagation (phase)**. The kernel-weighted update for token $i$ is

$$\mathfrak{h}'_i = \sum_{j=1}^{N} \kappa(\mathbf{h}_{P,i}, \mathbf{h}_{P,j}; \mathbf{H}_P) \cdot \big( \psi_A(\mathbf{h}_{A,j}), \psi_P(\mathbf{h}_{P,j}) \big), \tag{5}$$

which yields the matrix form

$$\mathbf{H}'_A = \boldsymbol{\alpha} \cdot \psi_A(\mathbf{H}_A), \tag{6}$$

$$\mathbf{H}'_P = \boldsymbol{\alpha} \cdot \psi_P(\mathbf{H}_P). \tag{7}$$

**Summary.** The **Propagation Kernel** computes interactions from **phase** alone; the **Dual-State Operator** then transports **energy** and **propagation** content along this field. This provides a compact, physics-aligned inductive bias beyond generic attention, while remaining implementation-compatible with standard attention kernels.

### 3.2.2 Scale-Conditional Projection for Dynamics Modeling

Following the global interaction captured by the propagation-kernel operator, the block must perform a local, token-wise transformation to model the intrinsic evolution of each spectral component. A standard, monolithic feed-forward network (FFN) is ill-suited for this task, as it would apply a uniform transformation across all physical scales, contrary to the heterogeneous nature of atmospheric dynamics. To overcome this, we introduce a **Scale-Conditional Projection** mechanism that allows the model to apply specialized, physically-aware updates based on each token's intrinsic scale.

**Evolutionary Subspace Projection.** We conceptualize the evolution of each dual state as a projection onto two distinct, learned manifolds, which we term "evolutionary subspaces." These are modeled by two specialized, non-linear operators:

- The **Energy Evolution Projector**, $\Phi_A : \mathbb{R}^{d_{\text{model}}} \to \mathbb{R}^{d_{\text{model}}}$, an expert network trained to model the dynamics of system intensification and decay (i.e., amplitude change).
- The **Propagation Evolution Projector**, $\Phi_P : \mathbb{R}^{d_{\text{model}}} \to \mathbb{R}^{d_{\text{model}}}$, an expert network trained to model the dynamics of spatial translation and rotation (i.e., phase change).

**The Adaptive Dynamics Arbiter (ADA).** Instead of applying these projections statically, the model must learn to dynamically arbitrate between them based on the current state and, crucially, its physical scale. For this, we design an **Adaptive Dynamics Arbiter (ADA)**, a gating function $g_k \in [0, 1]$ that determines the appropriate evolutionary path for each token $k$. The arbiter's decision is conditioned on the complete local state information: the intermediate energy and propagation states $(\mathbf{h}'_{A,k}, \mathbf{h}'_{P,k})$ and the static scale embedding $\mathbf{s}_k$:

$$g_k = \sigma \big( \text{Gate}_\theta \big( [\mathbf{h}'_{A,k}; \mathbf{h}'_{P,k}; \mathbf{s}_k] \big) \big) \tag{8}$$

where $\text{Gate}_\theta$ is a learned neural network and $\sigma$ is the sigmoid activation. This gating value $g_k$ can be interpreted as a learned physical parameter that steers the evolution of the system.

The final state updates are then formulated as a controlled, residual projection onto the appropriate evolutionary subspace:

$$\mathbf{h}^{\text{out}}_{A,k} = \text{LayerNorm} \big( \mathbf{h}'_{A,k} + g_k \cdot \Phi_A(\mathbf{h}'_{A,k}) \big) \tag{9}$$

$$\mathbf{h}^{\text{out}}_{P,k} = \text{LayerNorm} \big( \mathbf{h}'_{P,k} + (1 - g_k) \cdot \Phi_P(\mathbf{h}'_{P,k}) \big) \tag{10}$$

This mechanism provides a powerful inductive bias, enabling the model to learn flexible and physically-plausible rules. For instance, the arbiter can learn to allocate most of the model's capacity to the Energy Evolution Projector (high $g_k$) for high-frequency tokens representing fast-developing convective systems, while prioritizing the Propagation Evolution Projector (low $g_k$) for low-frequency tokens representing the steady advection of planetary waves. This moves beyond simple gating, creating a learned, scale-aware parameterization of the underlying physical dynamics.

## 3.3 TRAINING OBJECTIVE

After the final SAED-Former block, we map the dual states to the next-step *complex* spectrum by predicting amplitude and phase separately:

$$\hat{\mathbf{A}}_{t+1} = f_A(\mathbf{H}_A) \in \mathbb{R}_{\geq 0}^{H \times W}, \qquad \hat{\mathbf{P}}_{t+1} = f_P(\mathbf{H}_P) \in (-\pi, \pi]^{H \times W}, \tag{11}$$

$$\hat{\mathbf{F}}_{t+1} = \hat{\mathbf{A}}_{t+1} \odot \exp(i\,\hat{\mathbf{P}}_{t+1}), \qquad \hat{\mathbf{X}}_{t+1} = \mathcal{F}^{-1}(\hat{\mathbf{F}}_{t+1}), \tag{12}$$

where $f_A, f_P$ are lightweight heads (e.g., linear/MLP), $\odot$ denotes element-wise product, and $\mathcal{F}^{-1}$ is the inverse FFT.

**Primary loss in physical space**  To account for latitude-dependent cell area, we use a latitude-weighted MSE on the physical field. Let $w_j = \cos\phi_j$ be the weight for latitude row $j$ and $Z = W \sum_{j=1}^{H} w_j$ the normalization constant. The main loss is

$$\mathcal{L}_{\text{spatial}} = \frac{1}{Z} \sum_{j=1}^{H} \sum_{i=1}^{W} w_j \left((\mathbf{X}_{t+1})_{j,i} - (\hat{\mathbf{X}}_{t+1})_{j,i}\right)^2. \tag{13}$$

**Optional auxiliary loss in frequency space**  We optionally supervise amplitude and phase separately to encourage the dual-state factorization:

$$\mathcal{L}_A = \left\|\mathbf{A}_{t+1} - \hat{\mathbf{A}}_{t+1}\right\|_2^2, \quad \mathcal{L}_P = \left\|\text{wrap}\left(\mathbf{P}_{t+1} - \hat{\mathbf{P}}_{t+1}\right)\right\|_2^2 \tag{14}$$

with total frequency loss $\mathcal{L}_{\text{freq}} = \lambda_A \mathcal{L}_A + \lambda_P \mathcal{L}_P$. The operator $\text{wrap}(\cdot)$ applies element-wise angular wrapping to $(-\pi, \pi]$:

$$\text{wrap}(\Delta) = \left((\Delta + \pi) \bmod 2\pi\right) - \pi. \tag{15}$$

**Overall objective**  The final objective combines the physical-space error with the (optional) frequency-space supervision:

$$\mathcal{L}_{\text{total}} = \mathcal{L}_{\text{spatial}} + \mathcal{L}_{\text{freq}}. \tag{16}$$

## 4 EXPERIMENTS

In this section, we address six key research questions:

- **RQ1: Comparative skill** – Performance vs. SOTA across variables/horizons
- **RQ2: Efficiency** – Training costs (GPU memory, time) vs. baselines
- **RQ3: Transfer** – Zero/few-shot cross-variable prediction capability
- **RQ4: Ablations** – Marginal contribution of each component
- **RQ5: Sensitivity** – Hyperparameter impact analysis
- **RQ6: Qualitative** – Real-world extreme weather case studies

### 4.1 EXPERIMENTAL SETUP

**Datasets.** We use the **ERA5 reanalysis** dataset Hersbach et al. (2020); Rasp et al. (2024), the fifth-generation ECMWF global reanalysis. Experiments adopt the $5.625°$ version ($64 \times 32$ grid) from 2006–2018. We evaluate four variables: **2 m temperature (2mT)**, **10 m U-wind (u10)**, **geopotential (Z)**, and **temperature (T)** at pressure levels (details in Appendix B.1, Table 6).

**Evaluation Metrics.** Following Rasp et al. (2024), we report **RMSE** and **ACC**. To correct area distortion on a latitude–longitude grid, all metrics are latitude-weighted; precise formulas appear in Appendix B.5.

**Baselines.** We compare against state-of-the-art models under matched settings: **NODE** Chen et al. (2019), **FourCastNet** Pathak et al. (2022), **ClimaX** Nguyen et al. (2023), and **ClimODE** Verma et al. (2024). See Appendix B.2 for configurations.

**Parameter Settings.** ERA5 is split into train (2006–2015), val (2016), and test (2017–2018). Baselines use their reported best hyperparameters. We train with batch size 64, learning rate $1 \times 10^{-4}$, and the Adam optimizer Kingma & Ba (2017).

Table 1: Performance comparison of different models on weather forecasting tasks. The table shows ACC metrics across different variables and lead times.

| Model | z | | | | t | | | | t2m | | | |
|---|---|---|---|---|---|---|---|---|---|---|---|---|
| | 6h | 12h | 18h | 24h | 6h | 12h | 18h | 24h | 6h | 12h | 18h | 24h |
| NODE | 0.96 | 0.88 | 0.79 | 0.70 | 0.94 | 0.85 | 0.77 | 0.72 | 0.82 | 0.68 | 0.69 | 0.79 |
| ClimaX | 0.97 | 0.96 | 0.95 | 0.93 | 0.94 | 0.93 | 0.92 | 0.90 | 0.92 | 0.90 | 0.88 | 0.89 |
| ClimODE | 0.99 | 0.99 | 0.98 | 0.98 | 0.97 | 0.96 | 0.96 | 0.95 | 0.97 | 0.96 | 0.96 | 0.96 |
| FCN | 0.99 | 0.99 | 0.99 | 0.99 | 0.99 | 0.99 | 0.99 | **0.99** | 0.99 | 0.99 | 0.99 | 0.99 |
| ClimateLLM | **1.00** | **1.00** | **0.99** | **0.99** | **1.00** | **0.99** | **0.99** | 0.98 | **1.00** | **1.00** | **0.99** | **0.99** |

## 4.2 Overall Performance (RQ1)

**Anomaly Correlation Coefficient (ACC).** From Table 1, ClimateLLM delivers best or tied-best short-range skill (6–24 h) across z/t/t2m, with only minor degradation over time that is comparable to strong baselines. This pattern matches our design: the **dual-state decoupling** eases short-horizon alignment, while the **phase-centric propagation kernel** preserves coherent propagation up to day-1 ranges.

Table 2: Comparison of different models' RMSE metrics variables at lead times of 6 hours.

| Variable | RMSE(↓) | | | | |
|---|---|---|---|---|---|
| | NODE | ClimaX | FCN | ClimODE | ClimateLLM |
| z | 300.64 | 247.5 | 149.4 | **112.3** | 139.5 |
| t | 1.82 | 1.64 | 1.18 | 1.19 | **1.02** |
| t2m | 2.72 | 2.02 | 1.28 | 1.27 | **1.01** |
| u10 | 2.3 | 1.58 | 1.47 | 1.48 | **1.41** |

**Root Mean Square Error (RMSE, 6 h).** Table 2 shows ClimateLLM achieves the lowest RMSE on t, t2m, and u10, and is second on z. The gains on near-surface variables align with our **scale-aware** updates (SAEM), which adapt evolution across wave bands; the z result remains competitive without specialized tuning.

### 4.2.1 Long-term Weather Forecasting

At 72/144 h (Table 3), ClimateLLM opens clear margins over strong baselines on z/t/t2m and remains tied or leading on u10. To gauge scale, at 144 h the ACC improves by $\sim$**66%** on u10 and $\sim$**48%** on z *relative to the strongest baseline*, with t and t2m also showing **double-digit** gains. These longer-horizon benefits align with our **phase-centric kernel** (maintaining propagation coherence) and **scale-aware evolution** (specializing dynamics across bands),

Table 3: Longer lead time predictions.

| Variable | Lead-Time (hours) | ACC(↑) | | |
|---|---|---|---|---|
| | | ClimaX | ClimODE | ClimateLLM |
| z | 72 | 0.73 | 0.88 | **0.95** |
| | 144 | 0.58 | 0.61 | **0.90** |
| t | 72 | 0.76 | 0.85 | **0.95** |
| | 144 | 0.69 | 0.77 | **0.94** |
| t2m | 72 | 0.83 | 0.85 | **0.98** |
| | 144 | 0.83 | 0.79 | **0.96** |
| u10 | 72 | 0.45 | **0.66** | **0.66** |
| | 144 | 0.30 | 0.35 | **0.58** |

yielding higher medium-range skill without extra compute (see **RQ2**).

## 4.3 Model efficiency (RQ2)

**Efficiency.** From Table 4, ClimateLLM reduces **time/epoch** $212.76 \rightarrow$ **31.53** s (**85.2%** ↓), **GPU memory** $34,900 \rightarrow$ **5,000** MB (**85.7%** ↓), and **total time** $17.6 \rightarrow$ **0.26** h (**98.5%** ↓). These savings follow from the **phase-centric kernel** (one propagation-driven influence shared across states) and

**SAEM** (per-band shared projections instead of a monolithic FFN), with frequency-domain mixing reducing required depth. All comparisons were performed on the same Nvidia A100.

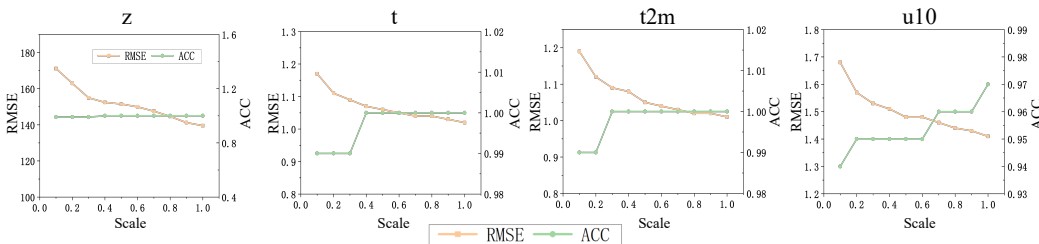

Figure 2: Few-shot Forecasting Results, with training sample scale ranging from 10% to 100%.

### 4.4 ZERO-SHOT AND FEW-SHOT FORECASTING (RQ3)

In our zero-shot and few-shot experiments (Shown in Fig. 2), ClimateLLM demonstrates exceptional generalization capabilities, achieving an accuracy of 0.99 in

Table 4: Efficiency Performance Comparison.

| Model | Training Time (s/epoch) | GPU Memory Usage (MB) | Total Training Time (hours) |
|---|---|---|---|
| ClimODE | 212.76 | 34,900 | 17.6 |
| ClimateLLM | **31.53** | **5000** | **0.26** |

zero-shot settings and outperforming baseline methods trained on full datasets with just **20%** of training data. For more detailed experimental results, please refer to Appendix D.

### 4.5 ABLATION EXPERIMENTAL STUDY (RQ4)

Table 5: Ablation study on key components of ClimateLLM.

| Model | z | | t | | t2m | | u10 | |
|---|---|---|---|---|---|---|---|---|
| | RMSE↓ | ACC↑ | RMSE↓ | ACC↑ | RMSE↓ | ACC↑ | RMSE↓ | ACC↑ |
| ClimateLLM | **139.5** | **1.00** | **1.02** | **1.00** | **1.01** | **1.00** | **1.41** | **0.97** |
| w/o FFT | 154.5 | 0.95 | 1.17 | 0.95 | 1.29 | 0.93 | 1.78 | 0.91 |
| w/o Dual-State | 143.7 | 0.98 | 1.08 | 0.98 | 1.09 | 0.99 | 1.57 | 0.94 |
| w/o Scale Embedding | 140.2 | 0.99 | 1.07 | 0.99 | 1.08 | 0.98 | 1.45 | 0.96 |

**Setup.** We ablate the three key components in Table 5: **FFT** (frequency-domain processing), **Dual-State** (amplitude/phase decoupling), and **Scale Embedding** (band-conditioned evolution via SAEM).

**Findings.** (1) **w/o FFT** yields the *largest* degradation across variables, consistent with our phase-centric design: removing the frequency representation eliminates the propagation field and band structure. For instance, ACC typically decreases by $0.05 \sim 0.07$ and RMSE increases markedly on **z** $(+15.0)$. (2) **w/o Dual-State** shows *moderate* and *consistent* drops (e.g., ACC $-0.01 \sim 0.03$), reflecting the cost of re-entangling **energy evolution (amplitude)** and **spatial propagation (phase)**. (3) **w/o Scale Embedding** produces *small but systematic* declines (ACC $-0.01 \sim 0.02$), with larger effects on near-surface, higher-wavenumber variables (t/t2m/u10), indicating that SAEM's band conditioning offers low-cost gains.

**Overall.** The contribution ranking **FFT** $\gg$ **Dual-State** $>$ **Scale Embedding** mirrors the model's inductive biases (spectral domain $\rightarrow$ dual-state factorization $\rightarrow$ scale-conditioned evolution), explaining why ClimateLLM maintains strong accuracy under compact compute.

## 4.6 SENSITIVITY ANALYSIS (RQ5)

Our sensitivity analysis reveals that ClimateLLM maintains consistent performance across different SAED-Former layers configurations (1, 2, 4, 6, and 8), with negligible variations in both RMSE and ACC metrics, indicating robust performance regardless of model depth. For detailed sensitivity analysis results, please refer to Appendix E.

## 4.7 EXTREME WEATHER CASE ANALYSIS (RQ6)

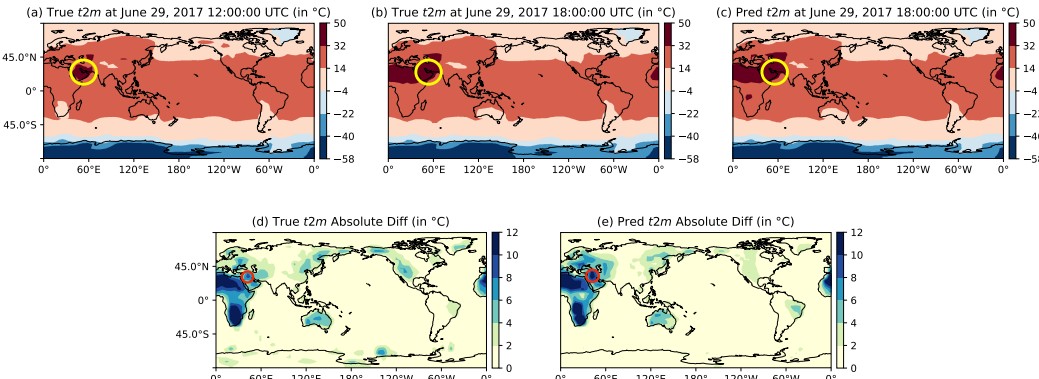

Figure 3: Case Study of variable $t2m$ on 2017-06-29. (a) True value at $t_0$ (b) True value at $t_1$ (c) ClimateLLM prediction results at $t_1$ (d) The difference between true value at $t_0$ and true value at $t_1$ (e) The difference between prediction result at $t_1$ and true value at $t_0$.

**Case Study: June 2017 Ahvaz Heat Event.** We examine a rapid near-surface warming over southwest Iran on **June 29, 2017** between **12:00** and **18:00** UTC, covering the widely reported Ahvaz heat episode Kapikian & Samenow (2017) (yellow circle in Fig. 3). Panels(a,b) show the observed $t2m$ fields and their 6 h change at $5.625°$ resolution; panel(c) is the **6 h ahead** forecast from ClimateLLM, with differential maps in (d,e). The prediction **co-locates** the warming maximum with the observed hotspot and **reproduces** the anisotropic warming gradient aligned with the low-level flow. Within the boxed Middle East region, the predicted sign and spatial extent of the heat closely follow the analysis, indicating **structure-preserving skill** in both *location* and *orientation*.

Methodologically, this outcome is consistent with our inductive biases: the **phase-centric propaga-tion kernel** maintains *coherent transport* of the warm anomaly, while the **Scale-Aware Evolution Module (SAEM)** specializes updates in higher wavenumber bands characteristic of $t2m$ variability. The **dual-state** design allows amplitude growth (energy evolution) to be modeled separately from phase-driven displacement (spatial propagation), which is critical for fast-onset heat events. All evaluations use ERA5 fields (units in °C) at $5.625°$ on a latitude–longitude grid; "change" denotes the difference between 18:00 and 12:00 UTC fields.

## 5 CONCLUSION

We introduced **ClimateLLM**, a frequency-domain foundation model for weather forecasting built on the **SAED-Former** backbone. By *decoupling* amplitude and phase into **dual states**, driving interactions with a **phase-centric propagation kernel**, and applying **scale-aware evolution** via band-conditioned updates, the model learns compact, physics-aligned representations. Across vari-ables and horizons, ClimateLLM matches or surpasses strong baselines while using substantially less compute; it also exhibits robust zero-/few-shot transfer and captures challenging events in case studies.

**Future work.** We plan to (i) incorporate *physics-guided constraints* (e.g., conservative spectral operators for mass/energy) and diagnostics; and (ii) scale to higher resolutions and regional special-izations while preserving efficiency.

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

## A  METHOD

The complete algorithm workflow is described in Algorithm 1.

---

**Algorithm 1** 2D FFT-based Climate State Processing Pipeline (SAED-Former)

---

**Input:** Climate states $\{X(l) \in \mathbb{R}^{|\mathcal{V}| \times M \times N}\}_{l=t-L+1}^{t}$, historical length $L$, inverse FFT $\mathcal{F}^{-1}(\cdot)$.
**Output:** Predicted climate state $X_{\text{pred}}(t+1) \in \mathbb{R}^{|\mathcal{V}| \times M \times N}$.

1: **Data Normalization:**
2: Compute per-variable mean and std over the context:

$$\mu(v) = \frac{1}{LMN} \sum_{l=t-L+1}^{t} \sum_{m=1}^{M} \sum_{n=1}^{N} X(l)[v,m,n], \quad \sigma^2(v) = \frac{1}{LMN} \sum_{l=t-L+1}^{t} \sum_{m,n} \big(X(l)[v,m,n] - \mu(v)\big)^2$$

3: Normalize each frame:

$$\hat{X}(l)[v,m,n] = \frac{X(l)[v,m,n] - \mu(v)}{\sigma(v) + \epsilon} \quad \text{for } l = t-L+1, \ldots, t$$

4: **Apply 2D FFT:**
5: Transform each normalized frame to frequency domain:

$$S(l) = \mathcal{F}\big(\hat{X}(l)\big), \qquad S(l)[v, k_m, k_n] = \sum_{m=1}^{M} \sum_{n=1}^{N} \hat{X}(l)[v,m,n] \, e^{-2\pi i \left(\frac{k_m m}{M} + \frac{k_n n}{N}\right)}$$

6: **Frequency Tokenization & Scale Embedding:**
7: For each wavenumber $(k_m, k_n)$, decompose $S(l)[v, k_m, k_n] = A(l)[v, k_m, k_n] \, e^{iP(l)[v,k_m,k_n]}$.
8: Map $(k_m, k_n)$ to band $b = \mathcal{B}(k_m, k_n)$ and fetch scale vector $\mathbf{s} = \mathbf{E}_{\text{scale}}[b]$.
9: Initialize dual states per token:

$$\mathbf{h}_A^{(0)}(l, k_m, k_n) = \text{proj}_A\big(A(l)[\cdot, k_m, k_n]\big), \quad \mathbf{h}_P^{(0)}(l, k_m, k_n) = \text{proj}_P\big(P(l)[\cdot, k_m, k_n]\big)$$

10: **Phase-Centric Propagation Kernel (per SAED block):**
11: Compute queries/keys from propagation state and the influence field:

$$\mathbf{q}_i = \phi_q\big(\mathbf{h}_P^{(\ell-1)}(t,i)\big), \ \mathbf{k}_j = \phi_k\big(\mathbf{h}_P^{(\ell-1)}(t,j)\big), \ \alpha_{ij} = \frac{\exp\big(\langle \mathbf{q}_i, \mathbf{k}_j \rangle / \sqrt{d_k}\big)}{\sum_l \exp\big(\langle \mathbf{q}_i, \mathbf{k}_l \rangle / \sqrt{d_k}\big)}$$

12: Aggregate values to update both states at the block pre-output:

$$\mathbf{H}_A' = \boldsymbol{\alpha} \cdot \psi_A\big(\mathbf{H}_A^{(\ell-1)}\big), \qquad \mathbf{H}_P' = \boldsymbol{\alpha} \cdot \psi_P\big(\mathbf{H}_P^{(\ell-1)}\big)$$

13: **Scale-Aware Evolution Module (SAEM):**
14: For each token $i$ with band $b_i$ and scale $\mathbf{s}_i$, apply band-conditioned projections and gates:

$$\tilde{\mathbf{h}}_A = \Phi_A^{(b_i)}\big(\mathbf{h}_{A,i}'\big), \quad \tilde{\mathbf{h}}_P = \Phi_P^{(b_i)}\big(\mathbf{h}_{P,i}'\big), \quad \boldsymbol{z}_A = \sigma\big(\mathbf{W}_A[\mathbf{h}_{A,i}'; \mathbf{h}_{P,i}'; \mathbf{s}_i]\big), \ \boldsymbol{z}_P = \sigma\big(\mathbf{W}_P[\mathbf{h}_{A,i}'; \mathbf{h}_{P,i}'; \mathbf{s}_i]\big)$$

15: Residual updates of dual states:

$$\mathbf{h}_{A,i}^{(\ell)} = \mathbf{h}_{A,i}' + \boldsymbol{z}_A \odot \tilde{\mathbf{h}}_A, \qquad \mathbf{h}_{P,i}^{(\ell)} = \mathbf{h}_{P,i}' + \boldsymbol{z}_P \odot \tilde{\mathbf{h}}_P$$

16: **Prediction Head & Inverse 2D FFT:**
17: Predict next-step amplitude/phase from final states at $t$:

$$\hat{A}(t+1) = f_A\big(\mathbf{H}_A^{(L)}(t)\big), \qquad \hat{P}(t+1) = f_P\big(\mathbf{H}_P^{(L)}(t)\big), \qquad \hat{S}(t+1) = \hat{A}(t+1) \odot e^{i\hat{P}(t+1)}$$

18: Reconstruct spatial field:

$$\hat{X}_{\text{pred}}(t+1) = \mathcal{F}^{-1}\big(\hat{S}(t+1)\big), \qquad X_{\text{pred}}(t+1) = R_{de}\big(\hat{X}_{\text{pred}}(t+1)\big)$$

---

Table 6: Variables of the ERA5 datasets.

| Variable name | Abbrev. | ECMWF ID | Levels | Units |
|---|---|---|---|---|
| 2 meter temperature | t2m | 167 | - | K |
| 10 meter U wind component | u10 | 165 | - | $\mathrm{m\,s^{-1}}$ |
| Geopotential | z | 129 | 500 | $\mathrm{m^2\,s^{-2}}$ |
| Temperature | t | 130 | 850 | K |

# B  EXPERIMENTAL SETTINGS

## B.1  DATASETS

We trained our model using the ERA5 datasets from WeatherBench2 Rasp et al. (2024). Weather-Bench 2 is a framework for evaluating and comparing data-driven and traditional numerical weather forecasting models. All data used in our experiments are available at: `https://github.com/google-research/weatherbench2`

## B.2  BASELINES

- **NODE** Chen et al. (2019): Neural Ordinary Differential Equations (NODE) model is a continuous-depth neural network model and uses differential equation solvers to compute outputs by parameterizing the derivatives of hidden states.

- **FourCastNet** Pathak et al. (2022): FourCastNet is a deep learning model developed for global weather forecasting that uses the Vision Transformer (ViT) and Fourier Neural Operator (FNO) architecture for weather prediction.

- **ClimaX** Nguyen et al. (2023): ClimaX is a foundation model using self-supervised learning for weather and climate science that uses a transformer-based architecture to handle multiple types of Earth system data.

- **ClimODE** Verma et al. (2024): ClimODE implements weather prediction as a physics-informed neural ODE based on the principle of advection. It models weather as a continuous-time transport process through a hybrid neural network combining local convolutions and global attention.

## B.3  GPT BACKBONE SETTINGS

In our implementation, we adopt the GPT backbone architecture from TEMPO Cao et al. (2023). The specific configuration details are as follows:

- **Base Model**: We utilize the pre-trained GPT-2 architecture as our foundation.

- **Model Size**: The configuration consists of 6 transformer layers with a hidden dimension size of 768 and 12 attention heads.

- **Sequence Length**: Maximum sequence length is set to 1024 tokens to effectively capture long-range dependencies in time series data.

- **Component Processing**: Following TEMPO's methodology, we process trend, seasonal, and residual components separately before feeding them into the transformer layers.

- **Normalization**: We implement reverse instance normalization on each global component and local input to facilitate knowledge transfer and minimize distribution shift losses.

- **Fine-tuning Approach**: We utilize Low-Rank Adaptation (LoRA) with a rank of 8 to efficiently adapt the pre-trained weights to time series forecasting tasks while minimizing the number of trainable parameters.

This architecture enables our model to effectively leverage the knowledge encoded in the pre-trained transformer while adapting it to the unique characteristics of time series data. The combination of decomposition-aware processing and prompt-based adaptation allows the model to handle different temporal patterns more effectively than standard transformer architectures applied directly to raw

time series data. For training, we use the Adam optimizer with a learning rate of $1 \times 10^{-4}$ and weight decay of $1 \times 10^{-5}$. We implement a cosine learning rate scheduler with 10% warm-up steps. All experiments were conducted on NVIDIA A100 GPUs with batch size 64.

### B.4 SOFTWARE AND HARDWARE

The model is implemented with PyTorch Paszke et al. (2019) and the whole model training and inference is conducted on a single 80GB Nvidia A100 GPU.

### B.5 METRICS

In this paper, we focus mainly on the precision of the prediction of weather variables. Following related work Rasp et al. (2024), there are two metrics to evaluate the prediction accuracy, namely Root mean squared error (RMSE) and Anomaly correlation coefficient (ACC). Due to the varying grid cell areas in the equiangular latitude-longitude grid system (where polar cells are smaller than equatorial cells), we apply area-weighted metrics across grid points to prevent polar bias. The latitude weights $\alpha(m)$ are defined as:

$$\alpha(m) = \frac{\cos(m)}{\sum_{m'} \cos(m')} \tag{17}$$

where $m$ represents the latitude index of the grid point, and $L$ represents the latitude-dependent weighting factor used to account for the varying grid cell areas.

- **Root mean squared error (RMSE)** The latitude-weighted RMSE for a forecast variable $v$ at forecast time-step $l$ is defined by the following equation, with the same latitude weighting factor given by Equation 18,

$$\text{RMSE}(v) = \sqrt{\frac{1}{MN} \sum_{m=1}^{M} \sum_{n=1}^{N} \alpha(m)(\mathbf{X}_{\text{pred}}(m, n) - \mathbf{X}_{\text{true}}(m, n))^2} \tag{18}$$

  where $\mathbf{X}_{\text{true/pred}}(m, n)$ represents the value of predicted (/true) variable $v$ at the location denoted by the grid co-ordinates $(m, n)$ at a forecast time-step.

- **Anomaly correlation coefficient (ACC)** The latitude weighted ACC for a forecast variable $v$ at forecast time-step $l$ is defined as follows:

$$\text{ACC}(v) = \frac{\sum_{m,n} L(m)\tilde{\mathbf{X}}_{\text{pred}}\tilde{\mathbf{X}}_{\text{true}}}{\sqrt{\sum_{m,n} L(m)\tilde{\mathbf{X}}_{\text{pred}}^2 \sum_{m,n} L(m)\tilde{\mathbf{X}}_{\text{true}}^2}} \tag{19}$$

  where $\tilde{\mathbf{X}}_{\text{pred/true}} = \mathbf{X}_{\text{pred/true}} - C$ represents the long-term-mean-subtracted value of predicted (/true) variable $v$. While $C = \frac{1}{N} \sum_{t}^{N} \mathbf{X}_{\text{true}}$ is the climatology mean of the history. For more detail, please refer to Appendix B.5.

## C INTERPRETING MODEL PREDICTIONS FROM FREQUENCY DOMAIN

**Proposition 1 (Equivalence of Time-Domain Forecasting and Frequency-Domain Forecasting for 2D FNO)**

*Assume $\{(x_0, y_0), (x_1, y_1), \ldots, (x_{N-1}, y_{N-1})\}$ is the input sequence in the time domain, and $\{(\hat{x}_0, \hat{y}_0), (\hat{x}_1, \hat{y}_1), \ldots, (\hat{x}_N, \hat{y}_N)\}$ is the predicted output sequence of the frequency model. The predicted value $(\hat{x}_N, \hat{y}_N)$ is obtained by transforming from the frequency domain to the time domain at timestamp $N$.*

*Proof.* Assume $\{(x_0, y_0), (x_1, y_1), \ldots, (x_{N-1}, y_{N-1})\}$ is the input sequence in the time domain, and $\{(\hat{x}_0, \hat{y}_0), (\hat{x}_1, \hat{y}_1), \ldots, (\hat{x}_N, \hat{y}_N)\}$ is the predicted output sequence of the frequency model. The predicted value $(\hat{x}_N, \hat{y}_N)$ is obtained by transforming from the frequency domain to the time

domain at timestamp $N$. In this context, the prediction of the next frequency component $F'(u, v)$ in the frequency domain allows for forecasting the next values in the time domain.

The 2D Discrete Fourier Transform (DFT) and its inverse (iDFT) are defined as:

$$F(u, v) = \frac{1}{N^2} \sum_{x=0}^{N-1} \sum_{y=0}^{N-1} f(x, y) e^{-\frac{2\pi i}{N}(ux+vy)}, \quad u, v = 0, 1, \ldots, N-1, \tag{20}$$

$$f(x, y) = \sum_{u=0}^{N-1} \sum_{v=0}^{N-1} F(u, v) e^{\frac{2\pi i}{N}(ux+vy)}, \quad x, y = 0, 1, \ldots, N-1. \tag{21}$$

We introduce coefficients $A$ and $B$ to describe the relationship between the known time-domain sequence and its frequency-domain representation:

$$A = \sum_{x=0}^{N-1} \sum_{y=0}^{N-1} f(x, y) \left( \frac{e^{-\frac{2\pi i}{N}(ux+vy)}}{N} - \frac{e^{-\frac{2\pi i}{N+1}(ux+vy)}}{N+1} \right), \tag{22}$$

$$B = \frac{1}{(N+1)^2} \sum_{x=0}^{N-1} \sum_{y=0}^{N-1} f(x, y) e^{-\frac{2\pi i}{N+1}(ux+vy)}. \tag{23}$$

The new time-domain values $f(N, y)$ and $f(x, N)$ can be predicted as:

$$f(N, y) = (N+1) \left( F'(N, y) - B \right) e^{-\frac{2\pi i}{N+1}N^2}, \tag{24}$$

$$f(x, N) = (N+1) \left( F'(x, N) - B \right) e^{-\frac{2\pi i}{N+1}N^2}. \tag{25}$$

Similarly, the new frequency-domain values $F'(u, v)$ are given by:

$$F'(u, v) = A + (F(N+1, v) - B) e^{\frac{2\pi i}{N+1}(ux+vy)}, \quad u, v = 0, 1, \ldots, N-1. \tag{26}$$

Thus, for each $u, v$, the new frequency component $F'(u, v)$ can be inferred from the relationship:

$$F'(u, v) = A + (F'(u, v) - B) e^{\frac{2\pi i}{N+1}(ux+vy)}. \tag{27}$$

Once $F'(u, v)$ is determined, the predicted time-domain values $f(N, y)$ and $f(x, N)$ can be obtained by applying the inverse 2D DFT in equation 21.

In conclusion, the 2D FNO predicts the next frequency component $F'(u, v)$ by using the relationship between time-domain and frequency-domain representations. The coefficients $A$ and $B$ are used to infer the new frequency-domain values from the known values $F(u, v)$. Finally, the inverse DFT transforms $F'(u, v)$ back to the time domain to obtain the predicted value $(\hat{x}_N, \hat{y}_N)$. $\square$

## D   ZERO-SHOT AND FEW-SHOT FORECASTING (RQ3)

Table 7: Zero-shot Forecasting Results. Left of the arrow $\rightarrow$ training samples, right $\rightarrow$ test samples.

| Prediction Task | Metric | Value |
|---|---|---|
| $t \rightarrow t2m$ | RMSE | 2.01 |
| | ACC | 0.99 |
| $t2m \rightarrow t$ | RMSE | 1.21 |
| | ACC | 0.99 |

To assess foundation-model generalization, we evaluate **zero-shot** and **few-shot** settings. In the zero-shot protocol (no target-variable finetuning), Table 7 shows strong transfer on **t** and **t2m**: ACC = 0.99 for both, exceeding full-shot **ClimODE** (0.97/0.96). For **t**, this comes with a comparable RMSE (1.21 vs. 1.19). In the few-shot regime (Figure 2), using only **20%** of training data already yields **ACC ≈ 0.99** on **z/t/t2m**, surpassing **ClimODE/ClimaX** trained on the full dataset, with consistently lower RMSE than baselines at the same data fraction.

*Why does this hold?* The dual-state factorization learns variable-agnostic *propagation* priors in the phase pathway and *energy* priors in the amplitude pathway, reducing the need for variable-specific statistics. The **phase-centric propagation kernel** transports information along physically coherent pathways that are shared across variables (e.g., advection), while the **scale-aware evolution (SAEM)** specializes dynamics by wave band, supporting transfer from data-rich to data-scarce spectra. In few-shot, only lightweight heads (and limited normalization) need adaptation, so the frozen spectral backbone provides a strong inductive prior that converts scarce labels into rapid skill gains.

# E  SENSITIVITY ANALYSIS (RQ5)

The generative pre-trained transformer serves as the primary backbone of our ClimateLLM, and its parameter size often determines the model's representation capability at different levels. Therefore, in this section, we mainly analyze the sensitivity of the number of SAED-Former layers. As demonstrated in Figure 4, our experimental results reveal that varying the number of SAED-Former layers (1, 2, 4, 6, and 8) produced negligible differences in both RMSE and ACC metrics across variables, suggesting that our model demonstrates low sensitivity to the quantity of SAED-Former layers.

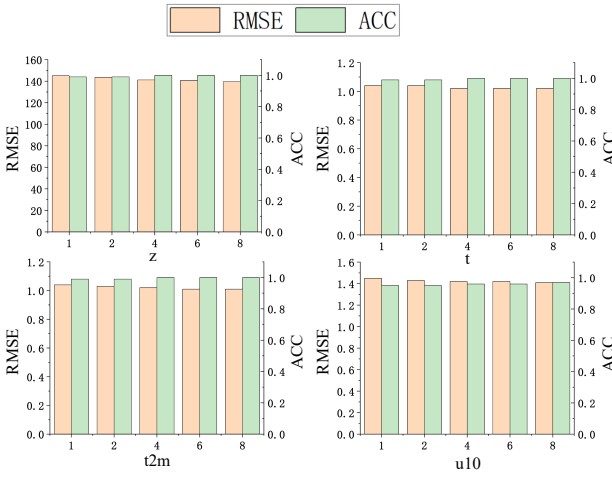

Figure 4: Sensitivity analysis of SAED-Former's number of layers.

# F  EXTRA CASE STUDY RESULT

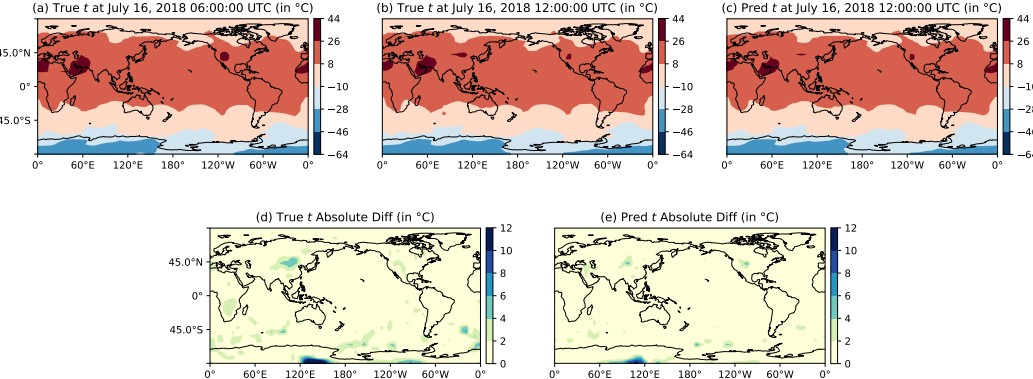

Figure 5: Case Study of variable t

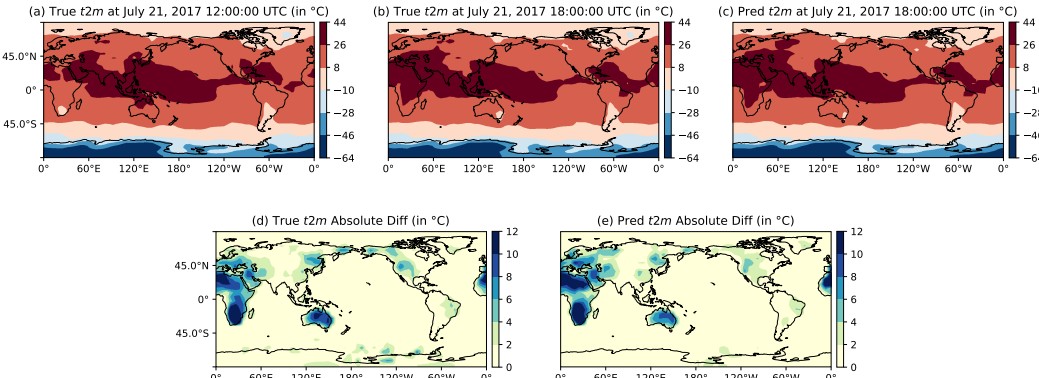

Figure 6: Case Study of variable t2m

Here in Figure 5 and Figure 6 we present another case study focusing on the variable t and t2m, examining the time period from July 16, 2018 06:00:00 UTC to July 16, 2018 12:00:00 UTC. Our model demonstrates comparable efficacy in capturing these dramatic weather transitions, further validating its robust performance in detecting significant meteorological variations.

## G   DECLARATION ON THE USE OF LARGE LANGUAGE MODELS

We employed GPT-2 as a backbone component in our proposed architecture (Fig. 1). Separately, we used large language models (e.g., GPT-5 and related GPT systems) to assist with copy-editing and grammar refinement. All model-assisted outputs were reviewed, revised, and validated by the authors. The authors assume full responsibility for the final content.

