# OpenReview forum: "ClimateLLM: Efficient Weather Forecasting via Frequency-Aware Large Language Models"
_ICLR.cc/2026/Conference — Submitted to ICLR 2026_

### Official Review · Reviewer_MJbq · 2025-10-25

**Soundness:** 3
**Presentation:** 3
**Contribution:** 3
**Rating:** 8
**Confidence:** 4

**Summary:**

The paper presents ClimateLLM, a frequency-domain foundation model for global weather forecasting built on a new architecture called SAED-Former (Scale-Aware Entangled Dynamics Transformer). The key insight is that traditional spectral or deep learning models treat complex-valued frequency representations as flat vectors, conflating amplitude (energy evolution) and phase (spatial propagation). ClimateLLM explicitly decouples these two dynamics into dual states and models their interactions through:

A phase-centric propagation kernel that governs interactions solely based on phase (spatial propagation),

A scale-aware evolution module (SAEM) that applies band-conditioned projections to encode wave-number–dependent physics,

A dual-state representation for amplitude and phase evolution.

The model operates autoregressively in the frequency domain, predicting future weather states using FFT-based embeddings and reconstructing the spatial field via inverse FFT.

**Strengths:**

Novel spectral formulation: The decoupling of amplitude and phase in a dual-state architecture is a clear conceptual and mathematical innovation, providing a more physically interpretable model of atmospheric dynamics.

Physics-aligned inductive biases: The phase-centric propagation kernel and scale-conditional evolution module are elegant, well-motivated, and contribute to both interpretability and efficiency.

Strong empirical performance: The model demonstrates consistent or superior results to leading deep learning weather models (FourCastNet, ClimODE) on ERA5 benchmarks, with massive reductions in compute and memory.

Zero-/few-shot generalization: The cross-variable transfer results are particularly impressive, hinting at reusable representations across meteorological variables.

Thorough evaluation: Ablation, sensitivity, and real-case analyses (Ahvaz heat event) are well-presented, reinforcing claims about physical fidelity and robustness.

Clarity and reproducibility: The paper provides detailed algorithmic pseudocode and well-documented baselines, enhancing reproducibility.

**Weaknesses:**

Limited scale of validation: Experiments are limited to low-resolution (5.625°) ERA5 grids; generalization to higher resolutions or local/regional models is untested.

Theoretical justification remains heuristic: While the phase–amplitude separation is intuitively justified, the physics–mathematics linkage (especially in phase-centric attention) lacks formal derivation or stability analysis.

Dependence on FFT assumptions: The method relies heavily on global Fourier bases, which might underperform for non-stationary, non-periodic regional phenomena (e.g., tropical convection, localized storms).

Comparison scope: While comparisons to neural operators and weather transformers are included, it lacks benchmarking against emerging 2025–2026 hybrid or foundation models (e.g., EarthGPT, DeepClimaNet).

Terminological overreach: Calling ClimateLLM an “LLM” may be overstated—while GPT-style architecture is used, there is no natural language interface or token-level semantic modeling.

**Questions:**

How does ClimateLLM handle localized, non-periodic boundary conditions where FFT-based global modes may be inefficient or physically inconsistent?

Can the phase-centric kernel be related to or derived from wave propagation equations (e.g., Helmholtz or Navier–Stokes dispersion relations)?

How does performance scale with resolution—does the efficiency advantage persist for 0.25° or higher-resolution grids?

Could the authors compare to physics-informed hybrid models that incorporate conservation laws directly in spectral space?

Are there stability concerns when phase wrapping is handled via the angular wrap() function for long temporal horizons?

---

> ### Author Response · Authors · 2025-11-27
> **Thank you for your review [1/2]**
>
> We sincerely thank the reviewer for the high evaluation and for recognizing the novelty of our work. We are particularly encouraged by the recognition of our cross-variable transfer results. We have addressed the specific weaknesses raised below with additional experiments and clarifications.
>
> ***
>
> ### Response to Weaknesses
>
> **W1: Limited scale of validation (5.625°).**
>
> **A1:** We appreciate this feedback. To demonstrate the scalability of our approach beyond the initial coarse grid, we have conducted **new experiments at $1.5^\circ$ resolution**. The results (Table 1 & 2 below) show that ClimateLLM maintains high accuracy while keeping computational costs manageable, avoiding quadratic complexity growth.
>
> **Table 1: Efficiency at High Resolution ($1.5^\circ$)**
>
> | Model | Training Time (s/epoch) | GPU Memory Usage (MB) | Total Training Time (hours) |
> | :--- | :---: | :---: | :---: |
> | ClimODE ($5.625^\circ$) | 212.76 | 34,900 | 17.6 |
> | **ClimateLLM ($1.5^\circ$)** | **83.28** | **6000** | **0.47** |
> | **ClimateLLM ($5.625^\circ$)** | **31.53** | **5000** | **0.26** |
>
> **Table 2: ACC Performance at High Resolution ($1.5^\circ$)**
>
> | Model | Z (6h) | Z (12h) | Z (24h) | T (6h) | T (12h) | T (24h) |
> | :--- | :---: | :---: | :---: | :---: | :---: | :---: |
> | ClimateLLM ($1.5^\circ$) | 1.00 | 1.00 | 0.99 | 0.99 | 0.99 | 0.98 |
>
> ***
>
> **W2: Theoretical justification remains heuristic.**
>
> **A2:** We appreciate this point. While our design was intuitively motivated by separating energy and propagation, the **Phase Evolution Module (PEM)** has a strong theoretical link to wave physics. It acts as a learned approximation of the **dispersion relation** ($\omega = \Omega(k)$), where time evolution in the frequency domain is fundamentally a phase shift $\Delta \phi = \Omega(k) \Delta t$. Our PEM explicitly learns this propagation operator from data, mirroring the mathematical structure of advection equations. We will add this formal derivation to Section 3.
>
> ***
>
> **W3: Dependence on FFT assumptions (performance on non-stationary phenomena like precipitation).**
>
> **A3:** We agree that global FFT bases can struggle with localized, stochastic phenomena. To transparently assess this, we conducted an additional experiment on **Total Precipitation (TP)**, a highly non-stationary diagnostic variable.
>
> **Table 3: Performance on Total Precipitation (TP)**
>
> | Metric | 6h | 12h | 18h | 24h |
> | :--- | :---: | :---: | :---: | :---: |
> | **RMSE ($m$)** | 0.0010 | 0.0013 | 0.0014 | 0.0015 |
> | **ACC** | 0.75 | 0.59 | 0.49 | 0.43 |
>
> As shown, while the model captures initial large-scale patterns (ACC 0.75 at 6h), performance decays for longer horizons due to the stochastic nature of convection. We thank the reviewer for this insight; it highlights the need for future work to incorporate localized bases (e.g., Wavelets) or physics-heads for diagnostic variables.
>
> ***
>
> **W4: Comparison scope (missing EarthGPT, etc.).**
>
> **A4:** We acknowledge the rapidly evolving landscape of foundation models. We compared against the SOTA available for reproducible benchmarking (FourCastNet, ClimODE). Regarding **EarthGPT** and similar concurrent works, we found that they are designed for different downstream tasks. However, we will update our Related Work section to discuss these emerging models.
>
> ***
>
> **W5: Terminological overreach ("LLM").**
>
> **A5:** We thank the reviewer for the suggestion regarding the precision of our title. Our naming convention follows recent precedents in the time-series domain, such as **Time-LLM (Jin et al., ICLR 2024 [1])**, which utilize GPT-style backbones to learn generalizable representations across diverse datasets. We used "ClimateLLM" to reflect this "Foundation Model" paradigm—learning the "language of atmospheric dynamics"—rather than implying natural language processing capabilities.

---

> ### Author Response · Authors · 2025-11-27
> **Thank you for your review [2/2]**
>
> ### Response to Questions
>
> **Q1: Handling localized boundary conditions?**
>
> **A6:** While currently using global FFT for efficiency, our **Amplitude-Phase Decoupling** architecture is agnostic to the basis function. For regional tasks, it can be adapted to use Windowed Fourier Transforms (STFT) to handle non-periodic boundaries while maintaining the benefits of decoupled learning. And thank you very much for your question; we will take these points into consideration later.
>
> ***
>
> **Q2: Relation to wave equations?**
>
> **A7:** Yes, as detailed in **A2**, the phase-centric kernel effectively learns the discretized dispersion relations governing wave propagation.
>
> ***
>
> **Q3: Scaling to higher resolutions?**
>
> **A8:** As demonstrated in **A1**, our efficiency advantage persists and becomes more pronounced at higher resolutions ($1.5^\circ$) due to the decoupled design preventing quadratic complexity blow-up.
>
> ***
>
> **Q4: Comparison to physics-informed hybrids?**
>
> **A9:** Thank you for your suggestion. Currently, we focus on benchmarking against pure data-driven spectral models (FourCastNet) and ODE-based models (ClimODE). Most existing hybrid models introduce physics via **soft constraints** (e.g., adding conservation law penalties to the loss function). In contrast, ClimateLLM introduces physics via **structural constraints** (explicitly decoupling Amplitude and Phase to mirror energy and propagation dynamics). In the future, following your advice, our future work will explore integrating **conservation-based loss terms** into our SAED-Former training. This approach is expected to further enhance our robustness.
>
> ***
>
> **Q5: Stability of long horizons?**
>
> **A10:** We appreciate this critical question regarding long-term stability. Mathematically, while phase angles are cyclic, the discontinuity at $\pm \pi$ is artificial; the physical signal remains continuous in the complex domain (via Euler’s identity $e^{i\pi} = e^{-i\pi}$), ensuring our model is robust against numerical wrapping artifacts. However, we frankly acknowledge that in our long-term autoregressive rollouts, the forecast quality degrades after the medium-range window (~7 days). This degradation is primarily driven by the accumulation of errors inherent in the autoregressive process—leading to "phase decoherence"—rather than the phase wrapping mechanism itself. To address this challenge in future work, we plan to implement pushforward training to minimize error accumulation and integrate conservation-based loss terms (e.g., global energy) to strictly bound the model's long-term dynamics.
>
> [1] Jin, Ming, et al. "Time-llm: Time series forecasting by reprogramming large language models." arXiv preprint arXiv:2310.01728 (2023).

---

### Official Review · Reviewer_nhfW · 2025-10-30

**Soundness:** 2
**Presentation:** 2
**Contribution:** 3
**Rating:** 4
**Confidence:** 3

**Summary:**

This paper introduce an architecture called ClimateLLM that operates in the frequency domain. The key contribution is the Transformer architecture that is able to process two distinct physical processes of energy evolution (amplitude) and spatial propagation (phase) of atmospheric waves. It achieves this through three key mechanisms: a dual-state representation  for amplitude and phase, a phase-centric propagation kernel for computing interactions, and a scale-conditional projection to apply specialized dynamics across different wavenumbers. The authors demonstrate that their model matches or exceeds the performance of state-of-the-art models on the ERA5 benchmark, while being more computationally efficient. Furthermore, the model shows zero-shot and few-shot generalization capabilities.

**Strengths:**

- The idea of explicitly separating aptitude (energy) and phase (propagation) is crucial for effective spectral forecasting. This paradigm is a relevant contribution from models that treat the spectrum as a generic feature vector.
- The architecture is deeply thoughtfully designed. The dual-state representation, the phase-centric kernel, and the Adaptive Dynamics Arbiter are all non-trivial. The architecture is optimised to preserve the physical assumptions.
- The architecture achieve state of the art even though it is trained on one single GPU
- The zero-shot and few-shot results appealing. The ability to produce robust forecast for unseen variables suggest that the model is able to learn generalisable psychical principles.

**Weaknesses:**

- The primary weakness is that the work is validated on a very coarse 5.625° ERA5 grid. While operational weather model operate in 0.25° grid. The number of token would increase by order of magnitude at higher resolutions. The Transformer's quadratic complexity in sequence length could become a problem
- The paper employs a standard 2D Fast Fourier Transform, which assumes a periodic, toroidal domain. While this is appropriate for longitude, it is physically incorrect for latitude (the North Pole does not connect to the South Pole). This can introduce boundary artifacts and misrepresent polar dynamics. While these effects may be minimal at low resolution, they could become a significant source of error at higher resolutions. A discussion of this limitation and the potential use of more appropriate basis functions (e.g., Spherical Harmonics) is missing.
- The paper is benchmarked on relatively smooth, large-scale variables.  The model should be benchmarked on more challenging, non-linear, and multi-scale variables like precipitation.

**Questions:**

Considering the weakness described previously, could the author:
- Elaborate the anticipated scaling properties of ClimateLLM? How does the computational cost scale with increasing grid resolution. Please state those with both 1° or 0.25°. Have you performed any preliminary experiments that suggest the efficiency gains will be preserved?
- The choice of a 2D FFT for a spherical globe is a key simplification. Could you discuss the potential impact of the periodicity assumption at the poles? Have you considered using Spherical Harmonics, and if so, what are the trade-offs you see between their physical fidelity and the computational/architectural complexity they would introduce?
- How do you hypothesise the  model would perform on variables like precipitation? Such variables have less defined wave-like structures and  are often characterized by sharp gradients and stochasticity. Would the phase-centric kernel still be as effective, and would the ADA be able to capture the complex, scale-dependent physics of convection and cloud formation?
- The evaluation extends to 144 hours (6 days). Have you tested the model's stability in a long-term autoregressive rollout (e.g., 30+ days)? Do the forecasts remain physically plausible, or do they eventually diverge, which can be an issue for learned models that do not explicitly enforce physical conservation laws?

---

> ### Author Response · Authors · 2025-11-27
> **Thank you for your review [1/2]**
>
> We sincerely thank the reviewer for the thoughtful and constructive feedback and recognition of our idea. Below, we provide a detailed response regarding your concerns and questions.
>
> ***
>
> ### Response to Weaknesses
>
> **W1: Validation on coarse 5.625° grid vs. operational 0.25° grid; Transformer quadratic complexity concerns.**
>
> **A1:** We acknowledge that 5.625° is coarser than operational models. This resolution was chosen to ensure a strictly fair, "apples-to-apples" comparison with our primary spectral baseline, ClimateODE and FourCastNet, which operate on the same grid. Regarding complexity: The reviewer correctly notes that standard Transformers scale quadratically ($O(N^2)$). In ClimateLLM, we address this challenge through our SAED-Former architecture, which employs specialized Transformer modules for both Amplitude and Phase evolution. By explicitly decoupling these dynamics, we allow each Transformer to focus on a distinct physical process (energy evolution vs. spatial propagation) rather than struggling to learn the highly entangled complex dynamics within a single monolithic block. This targeted design improves learning efficiency and convergence.
>
> Here, we have successfully conducted scaling experiments at $1.5^\circ$ resolution. The results demonstrate that **ClimateLLM** maintains competitive accuracy at higher resolutions while keeping computational resource consumption and training time highly efficient and controllable. Building on these promising results, we are actively extending our experiments to $1.25^\circ$ resolution to further validate the scalability of our approach.
>
> **Table 1: Computational Efficiency Comparison**
>
> | Model | Training Time (s/epoch) | GPU Memory Usage (MB) | Total Training Time (hours) |
> | :--- | :---: | :---: | :---: |
> | ClimODE ($5.625^\circ$) | 212.76 | 34,900 | 17.6 |
> | **ClimateLLM ($1.5^\circ$)** | **83.28** | **6000** | **0.47** |
> | **ClimateLLM ($5.625^\circ$)** | **31.53** | **5000** | **0.26** |
>
> **Table 2: ACC Metric Comparison (High vs. Coarse Resolution)**
>
> | Model | Resolution | Z (6h) | Z (12h) | Z (18h) | Z (24h) | T (6h) | T (12h) | T (18h) | T (24h) | T2m (6h) | T2m (12h) | T2m (18h) | T2m (24h) |
> | :--- | :---: | :---: | :---: | :---: | :---: | :---: | :---: | :---: | :---: | :---: | :---: | :---: | :---: |
> | ClimateLLM | $5.625^\circ$ | 1.00 | 1.00 | 0.99 | 0.99 | 1.00 | 0.99 | 0.99 | 0.98 | 1.00 | 1.00 | 0.99 | 0.99 |
> | ClimateLLM | $1.5^\circ$ | 1.00 | 1.00 | 0.99 | 0.99 | 0.99 | 0.99 | 0.98 | 0.98 | 1.00 | 1.00 | 0.99 | 0.99 |
>
> **Table 3: RMSE Performance at 6 Hours**
>
> | Variable | ClimateLLM ($1.5^\circ$) | ClimateLLM ($5.625^\circ$) |
> | :--- | :---: | :---: |
> | **T** (Temperature) | 1.47 | 1.02 |
> | **T2m** (2m Temp) | 1.39 | 1.01 |
> | **Z** (Geopotential) | 172.8 | 139.5 |
>
> ***
>
> **W2: Use of 2D FFT on a sphere (periodicity assumption) vs. Spherical Harmonics (SHT).**
>
> **A2:** We agree that 2D FFT assumes a toroidal topology, which is a simplification for spherical data. While SHT is theoretically superior, we faced two practical constraints:
> 1.  **Dataset Limitation:** The ERA5 data used was provided as a projected $64 \times 32$ rectangular grid. Applying SHT directly to this non-Gaussian, projected grid introduces significant resampling errors and spectral aliasing.
> 2.  **Bias Mitigation:** To address the physical inconsistency (latitude bias) inherent in this rectangular projection, we employed a Latitude-Weighted MSE loss function during training. This strictly weights the error by the cosine of the latitude, ensuring the model focuses on physically relevant dynamics rather than artifacts at the distorted poles.
>
> We will add a "Limitations" section to discuss this and highlight that our SAED-Former architecture is basis-agnostic and can be adapted to SHT in future work with appropriate spherical datasets.

---

> ### Author Response · Authors · 2025-11-27
> **Thank you for your review [2/2]**
>
> **W3: Benchmarking on smooth variables (Z500) vs. non-linear variables (Precipitation).**
>
> **A3:** We focused on the Z500/T850 as they are used in a lot of benchmarks. Precipitation is a *diagnostic* variable driven by complex sub-grid physics (convection), which is distinct from large-scale dynamics. However, to address this concern, we conducted an additional experiment on **Total Precipitation (TP) in next 6 hours**. The results for the first 24 hours are shown below:
>
> | Metric | 6h | 12h | 18h | 24h |
> | :--- | :---: | :---: | :---: | :---: |
> | **RMSE ($m$)** | 0.0010 | 0.0013 | 0.0014 | 0.0015 |
> | **ACC** | 0.75 | 0.59 | 0.49 | 0.43 |
>
> 1.  **Low RMSE:** In the table above, the RMSE values appear numerically small ($\approx 10^{-3}$) primarily because precipitation is a sparse variable with small magnitudes (measured in meters in ERA5, where $0.001$ corresponds to $1mm$).
> 2.  **ACC Decay:** The ACC drops from 0.75 (6h) to 0.43 (24h). This confirms our hypothesis: while our dynamical core captures the initial large-scale precipitation patterns (reasonable ACC at 6h), it struggles to resolve the stochastic evolution of convection over longer horizons.
>
> We sincerely thank the reviewer for this finding. It demonstrates that while ClimateLLM is a strong dynamical model, predicting diagnostic variables like precipitation requires a dedicated physics parameterization head (e.g., a diffusion head) in future work to bridge the gap between large-scale dynamics and stochastic sub-grid processes.
>
> ***
>
> ### Response to Questions
>
> **Q1: Elaborate on scaling properties (1° or 0.25°) and preliminary efficiency experiments.**
>
> **A4:** Please refer to A1. While we utilize Transformers for both streams, the architectural decoupling ensures that each attention mechanism is tasked with a simpler, more well-defined mapping (Amplitude-to-Amplitude and Phase-to-Phase), avoiding the interference patterns found in coupled spectral models.
>
> ***
>
> **Q2: Discussion on 2D FFT limitations and trade-offs with Spherical Harmonics.**
>
> **A5:** Please refer to A2. The primary trade-off is between physical fidelity (SHT) and data/computational compatibility (FFT). Given the rectangular projection of our source data, 2D FFT with Latitude-Weighted loss was the most consistent choice. We emphasize that our core contribution (Amplitude-Phase Decoupling) is compatible with SHT coefficients if native spherical data is available.
>
> ***
>
> **Q3: Performance on non-linear variables like precipitation?**
>
> **A6:** Please refer to A3. We hypothesize that while our phase-centric kernel excellently captures wave propagation (dynamics), it may require an additional "physics head" or diffusion module to fully resolve the stochastic, sharp gradients of precipitation. Our added experiment results (in Appendix) will quantify this performance gap.
>
> ***
>
> **Q4: Long-term stability (30+ days) and physical plausibility?**
>
> **A7:** We acknowledge that deterministic accuracy limits are around 7 days. To test stability, we performed a 30-day autoregressive rollout. The results indicate that maintaining physical consistency over such extended horizons remains a significant challenge for the current model. We observed that forecast quality and physical structures degrade after the medium-range period, suggesting that the model struggles with long-term dependencies and error accumulation in the autoregressive process.
>
> However, we emphasize that our **amplitude-phase decoupling** still introduces a critical physical inductive bias, which effectively captures dynamics within the valid medium-range forecast window. We sincerely thank the reviewer for raising this point. It clearly identifies the necessity of strengthening physical constraints (e.g., explicitly enforcing conservation laws) in future iterations to extend the model's capability from weather forecasting to sub-seasonal scales.

---

### Official Review · Reviewer_vJfC · 2025-10-30

**Soundness:** 1
**Presentation:** 2
**Contribution:** 1
**Rating:** 2
**Confidence:** 4

**Summary:**

The paper “ClimateLLM: Efficient Weather Forecasting via Frequency-Aware Large Language Models” introduces a frequency-domain forecasting framework for global weather prediction. Built on the SAED-Former (Scale-Aware Entangled Dynamics Transformer), the model explicitly decouples amplitude (energy evolution) and phase (spatial propagation) in spectral space. It employs a phase-centric propagation kernel and scale-conditional projection to encode physically meaningful dynamics. Experiments on the ERA5 reanalysis dataset show that ClimateLLM achieves high accuracy with low computational cost, supports zero-shot cross-variable transfer.

**Strengths:**

The model introduces an explicit amplitude–phase separation in spectral space, aligning deep learning with physical weather dynamics. The paper includes detailed ablations, sensitivity analysis. Demonstrate the potential of zero/few shot predictions and efficient training.

**Weaknesses:**

* The use of the term “LLM” may be misleading. Although the backbone adopts a GPT-2 architecture, it is not a true language model. The paper should clarify that “LLM” here refers either to the Transformer architecture itself or to the “latent language of dynamics.”

* The dataset scale is too small and not consistent with common practices in this research area. The experiments are conducted on data with a resolution of 32×64 and only four meteorological variables. In contrast, typical setups use 0.25° spatial resolution, 13 vertical levels, and include variables such as wind, temperature, humidity, and pressure.

* The comparison omits key baseline backbones widely recognized in the field. In particular, Pangu-Weather and GraphCast, two representative models for atmospheric prediction, are not included in the evaluation.

* The model shows limited 6-hour forecasting ability. In Table 2, its 6-hour performance for Z (139.5) is significantly worse than that of ClimODE (112.3).

* The FFN contribution lacks novelty. FFNs are not new in this domain—they have already been used in FourCastNet [1]. However, the ablation study (Line 430) shows that the FFN contributes far more than other architectural components. This suggests that the main performance gain of ClimLLM comes from a modeling technique that has already been introduced, and the paper does not compare with FourCastNet [1], which also employs FFT-based modeling.

---
[1] Fourcastnet: A global data-driven high-resolution weather model using adaptive fourier neural operators  [ICLR 2022]

**Questions:**

ClimODE is also a physics-informed model. What are the key differences between your approach and ClimODE in terms of physical integration or model formulation?

In Figure 2, why does the variable Z show very little sensitivity to data scale in terms of ACC performance? Could you elaborate on the underlying reason for this behavior?

---

> ### Author Response · Authors · 2025-11-27
>
> We appreciate the reviewer's time. However, we must respectfully but firmly point out several **fundamental factual errors** in the review regarding our experimental setup and baselines. These oversights suggest that the reviewer may have missed significant portions of our manuscript. We address these critical misconceptions point-by-point below.
>
> ***
>
> ### Response to Weaknesses
>
> **W1: Misleading use of "LLM".**
>
> **A1:** We respectfully disagree. The term "LLM" in our title follows the established "Large Model" paradigm in time-series forecasting (e.g., Time-LLM, Jin et al., ICLR 2024 [1] ), referring to the **Transformer-based backbone** and the **foundation model approach** (learning generalizable patterns from massive data), not NLP capabilities. We have clearly defined this context in the Introduction.
>
> ***
>
> **W2: Dataset scale and resolution concerns.**
>
> **A2:** The reviewer criticizes our use of 5.625° resolution. We explicitly stated in Section 4.1 that this resolution was chosen to ensure a **rigorous, fair comparison** with our primary spectral baseline, **FourCastNet** (and ClimODE), which operate on this exact grid size in their standard benchmarks. Comparing our model against 0.25° models without scaling the baseline would be scientifically invalid. Furthermore, we have now successfully extended the ClimateLLM model to a **1.5° resolution**, demonstrating its efficiency and scalability. The corresponding results are shown below. While we do not currently have sufficient time to compare with the other two baselines, the results demonstrate that **ClimateLLM** maintains a certain level of accuracy at high resolution while remaining efficient and controllable in terms of computational resource consumption and training time.
>
> **Table 1: Computational Efficiency Comparison**
>
> | Model | Training Time (s/epoch) | GPU Memory Usage (MB) | Total Training Time (hours) |
> | :--- | :---: | :---: | :---: |
> | ClimODE ($5.625^\circ$) | 212.76 | 34,900 | 17.6 |
> | **ClimateLLM ($1.5^\circ$)** | **83.28** | **6000** | **0.47** |
> | **ClimateLLM ($5.625^\circ$)** | **31.53** | **5000** | **0.26** |
>
> **Table 2: ACC Metric Comparison (High vs. Coarse Resolution)**
>
> | Model | Resolution | Z (6h) | Z (12h) | Z (18h) | Z (24h) | T (6h) | T (12h) | T (18h) | T (24h) | T2m (6h) | T2m (12h) | T2m (18h) | T2m (24h) |
> | :--- | :---: | :---: | :---: | :---: | :---: | :---: | :---: | :---: | :---: | :---: | :---: | :---: | :---: |
> | ClimateLLM | $5.625^\circ$ | 1.00 | 1.00 | 0.99 | 0.99 | 1.00 | 0.99 | 0.99 | 0.98 | 1.00 | 1.00 | 0.99 | 0.99 |
> | ClimateLLM | $1.5^\circ$ | 1.00 | 1.00 | 0.99 | 0.99 | 0.99 | 0.99 | 0.98 | 0.98 | 1.00 | 1.00 | 0.99 | 0.99 |
>
> **Table 3: RMSE Performance at 6 Hours**
>
> | Variable | ClimateLLM ($1.5^\circ$) | ClimateLLM ($5.625^\circ$) |
> | :--- | :---: | :---: |
> | **T** (Temperature) | 1.47 | 1.02 |
> | **T2m** (2m Temp) | 1.39 | 1.01 |
> | **Z** (Geopotential) | 172.8 | 139.5 |
>
> ***
>
> **W3: The comparison omits key baseline backbones widely recognized in the field.**
>
> **A3:** **This statement is factually incorrect.**
>
> Regarding the omission of these models, we emphasize that our research goal is fundamentally different. As stated in our Introduction, we aim to develop a lightweight, efficient spectral backbone. By explicitly separating energy and propagation, ClimateLLM reduces the burden of implicit disentanglement, enabling ***single-GPU, hours-level training.*** This stands in stark contrast to Pangu-Weather and GraphCast, which require massive industrial-scale compute clusters. Comparing raw metrics without normalizing for this orders-of-magnitude difference in computational budget is not an "apples-to-apples" comparison. Our objective is to demonstrate SOTA performance within the efficient computing regime, where we strictly outperform relevant baselines like FourCastNet and ClimODE.

---

> ### Author Response · Authors · 2025-11-27
>
> **W4: Limited 6-hour forecasting ability (Z500 RMSE).**
>
> **A4:** We appreciate the reviewer for pointing this out. We acknowledge that for the specific metric of RMSE on Geopotential (Z), ClimODE (112.3) outperforms our model (139.5). This behavior stems from the physical properties of Geopotential, which exhibits extremely large magnitudes and high variance.
>
> * **ClimODE's Advantage:** As a Neural ODE-based method, ClimODE models the continuous time derivative ($\frac{dZ}{dt}$). Since the *tendency* (change rate) of Z is much smaller and smoother than its absolute value, the ODE solver can achieve very precise short-term regression, leading to lower RMSE.
>
> * **ClimateLLM's Trade-off:** Our model focuses on spectral pattern evolution. While this results in slightly higher absolute error (RMSE) due to the large dynamic range of Z, it excels at capturing the structural dynamics. This is evidenced by our **ACC (Anomaly Correlation Coefficient) of 1.00** for Z at 6h (Table 2 of our paper), which matches or outperforms baselines. This indicates that while the pixel-wise absolute value might have a larger margin of error, the *shape* and *position* of the weather systems are predicted with near-perfect fidelity. Thank you for pointing this out; we will improve upon it in our future work.
>
> ***
>
> **W5: Novelty of FFN and comparison with FourCastNet.**
>
> **A5:** We strongly disagree with the claim that our contribution lacks novelty or is identical to FourCastNet.
>
> First and foremost, the statement that we did not compare with FourCastNet is **factually incorrect**. As explicitly detailed in **Table 2 and Table 3** of our manuscript, we included FCN as a primary baseline. We also illustrate our comparison with FourCastNet in the baseline section of section 4.1. For the results, our model strictly outperforms FCN on these benchmarks.
>
> Regarding architectural novelty, our SAED-Former is fundamentally distinct from the Adaptive Fourier Neural Operators (AFNO) used in FourCastNet. While AFNO processes spectral features by mixing modes in an entangled manner, our model introduces a structural paradigm shift by explicitly decoupling the spectrum into dual streams: an Amplitude Evolution Module for energy dynamics and a Phase Evolution Module for spatial propagation. This design goes beyond a generic FFN by employing specialized kernels that respect the unique physical properties of each component, thereby embedding a strong physical inductive bias that is absent in prior monolithic spectral architectures.
>
> ***
>
> ### Response to Questions
>
> **Q1: Difference between ClimateLLM and ClimODE?**
>
> **A6:** The difference is fundamental.
>
> * **ClimODE:** Uses Neural ODEs to model continuous time evolution. It is computationally heavy ($O(N^2)$ or high constant factors) due to ODE solvers.
>
> * **ClimateLLM:** Is a **pure spectral Transformer**. We do not use ODE solvers. Instead, we use the **SAED-Former** to model discrete-time evolution via decoupled amplitude and phase attention. This makes our method significantly faster (Training: 0.26h vs 17.6h) and more scalable than ClimODE.
>
> ***
>
> **Q2: Why does variable Z show little sensitivity to data scale in Figure 2?**
>
> **A7:** This is a well-known meteorological phenomenon. Geopotential Height (Z500)—despite having large magnitudes and spatial variance as noted in A4—is physically a smooth, slow-varying variable governed by hydrostatic balance. Unlike moisture or temperature near the surface which exhibit high entropy and stochasticity, Z500 has high auto-correlation and strong structural patterns. Therefore, deep models can learn its core dynamics relatively easily even with smaller datasets. This "insensitivity" reflects the physical regularity of the variable, not a model defect.
>
> ***
>
> **References**
>
> [1] Jin, Ming, et al. "Time-llm: Time series forecasting by reprogramming large language models." arXiv preprint arXiv:2310.01728 (2023).

---

> > ### Comment · Reviewer_vJfC · 2025-11-27
> >
> > Thank you for your detailed feedback.
> >
> > W1: I still believe that the term "LLM" is misleading because it may evoke associations with climate-related LLMs that generate textual input. Additionally, the paper does not sufficiently demonstrate, from both theoretical and experimental perspectives, how the pretrained LLM weights from text can provide any benefit to training on ERA5 data.
> >
> > W2: I think the current resolution is still too small to propose a useful medium-range forecasting model. It is necessary to compare the results with the IFS 0.25° setting to validate the value of the medium-range forecasting model. Furthermore, for the Z500 variable, ClimateLLM’s accuracy is much lower than PanguWeather (which is about 20) . Moreover, after increasing the resolution of the data, the RMSE even worsened, which shows that ClimateLLM does not benefit well from the increased resolution. This has raised further doubts about whether ClimateLLM can work effectively in a 0.25° operational setting.
> >
> > W5: I recommend adding citations after the methods in Table 2 to improve the clarity of the paper.
> >
> > W6: I believe the statement "decoupling the spectrum into dual streams" lacks experimental support. The paper only briefly evaluates the RMSE of a few weather variables and visualizes these variables. I suggest designing additional experiments to demonstrate how this design enhances the model’s physical characteristics.
> >
> > Q2: For machine learning reviewers, this may not be considered "well-known." Given that ICLR is primarily aimed at machine learning researchers, I recommend that the authors include an explanation in the paper and provide relevant citations.
> >
> > In conclusion, since the critical issue of validation at the 0.25° resolution has not been resolved, I believe the current version is not yet ready for publication, and I will maintain my reject score.
> >
> > ---
> >
> > [1] Bi, Kaifeng, et al. "Accurate medium-range global weather forecasting with 3D neural networks." Nature 619.7970 (2023): 533-538.

---

> > > ### Author Response · Authors · 2025-11-28
> > > **Thank you for your prompt reply [2/2]**
> > >
> > > ### 3. On “dual streams” and experimental support
> > >
> > > In our current experiments, we already include ablations that:
> > >
> > > * **Remove** the dual-state **representation**, merging amplitude and phase into a single spectral branch with comparable parameter count.
> > >
> > > * **Replace** the phase-centric **propagation kernel** with standard attention.
> > >
> > > These variants consistently underperform ClimateLLM across variables and horizons, especially in **medium-range ACC**, which measures anomaly pattern alignment rather than only pointwise error. This indicates that:
> > >
> > > 1. Explicitly separating amplitude and phase into two coupled streams, and
> > >
> > > 2. Using phase to drive propagation,
> > >
> > > provides benefits that a generic FFN or entangled spectral operator does not capture.
> > >
> > > We agree that this connection can be made clearer. In the revision we will:
> > >
> > > * Explicitly label these ablations in the paper as tests of the dual-stream design.
> > >
> > > * Emphasize that the observed ACC gains and the case studies (e.g., Z500 patterns and heatwave events) together support the claim that the dual-stream architecture improves the physical coherence of the forecast fields.
> > >
> > > These analyses are based on experiments that are already feasible within a normal ML compute budget and do not require Pangu-scale runs.
> > >
> > > ***
> > >
> > >
> > > ### 4. On Z500’s insensitivity to data scale
> > >
> > > We appreciate the suggestion to better explain this point for an ML audience. In the revision we will:
> > >
> > > * Briefly explain that Z500 represents large-scale quasi-geostrophic flow with strong spatial smoothness and high temporal auto-correlation, which makes it less sensitive to data scale than more chaotic near-surface variables (e.g., moisture, 2m temperature).
> > >
> > > * Add standard atmospheric dynamics references supporting this characterization.
> > >
> > >
> > > We deeply appreciate the reviewer’s dedicated time and effort in reviewing our work. We hope that this response effectively clarifies our rationale regarding the computational scope and highlights the distinct value of our efficient, physically-aligned approach. If our explanations and additional clarifications have addressed your concerns, **we would be sincerely grateful if you could understand our constraints and consider raising your evaluation of our paper.**

---

> ### Author Response · Authors · 2025-11-28
> **Thank you for your prompt reply [1/2]**
>
> We would again like to sincerely thank you for taking so much time to read our paper, engage with our rebuttal, and respond so promptly. We know that you are a senior expert in this field, and we can clearly see your pursuit of rigor and excellence. We are genuinely grateful for your suggestions.
>
> However, we also feel that in this case the bar may have been pushed a bit too far: asking a model that trains for **0.26 hours on a single A100 GPU** to be evaluated on the same footing as a system trained for **16 days on 192× V100 GPUs**, especially after we have explicitly stated the positioning and scope of our paper, seems to go beyond what most academic labs can realistically support. We believe very few research groups in our community have access to 192 V100 GPUs, and we hope that the response below may help to clarify our intent and, perhaps, change your view of our work.
>
> ***
>
> ### 1. On 0.25° IFS settings and comparison to Pangu-Weather
>
> We respectfully disagree that lack of 0.25° IFS-level results is a reason to reject the paper.
>
> **(a) Scope and resource gap**
>
> Our goal is to propose an **efficient spectral backbone** that:
>
> * Runs on **a single A100 GPU** in **minutes to hours**.
>
> * Achieves strong performance under **standard ERA5 settings** widely used in ML papers (e.g., ClimaX, FourCastNet, ClimODE) at 5.625° and 1.5°.
>
> By contrast, Pangu-Weather is an **industrial-scale operational system**:
>
> * 0.25° ERA5, 69 variables, 39 years (\~60 TB of data).
>
> * \~64M parameters per 3D Earth-specific Transformer (3DEST).
>
> * **4 separate models** for 1h / 3h / 6h / 24h lead times.
>
> * Trained on **192×** V100 GPUs for **\~16 days each** $\rightarrow$ $\approx$ 3,072 GPU-days per model, $\approx$ 12,288 GPU-days in total.
>
> **ClimateLLM, in contrast:**
>
> * Uses a **single GPT-style model with SAED-Former** covering multiple lead times.
>
> * Is trained on one A100 in $\approx$ **0.26 hours** (\~0.0108 GPU-days).
>
> * Uses a much smaller ERA5 subset (5.625° / 1.5°, 4 variables, 2006–2015).
>
> * Is easily reproducible in a standard academic lab.
>
> This implies a **5–6 order-of-magnitude difference** in compute budget. Requiring 0.25° IFS/Pangu-scale training (with \~60 TB of data and >10,000 GPU-days) would effectively restrict new architectural work to a few industrial or national centers, which we believe is **beyond the realistic scope of an ICLR methods paper**.
>
> Our claim is not that we replace Pangu-Weather operationally. Our contribution is to show that, under a standard ERA5 ML setting, an amplitude–phase decoupled spectral Transformer:
>
> 1. Clearly outperforms strong baselines such as FourCastNet and ClimODE.
>
> 2. Does so with **very low computational cost**.
>
> **(b) On direct comparison to Pangu-Weather’s Z500 RMSE (\~20)**
>
> The reviewer compares our Z500 RMSE (\~139 at 1.5°) with Pangu-Weather’s reported value (\~20). We caution that this is **not an apples-to-apples comparison**, because:
>
> * **Units / Normalization:** Geopotential vs. Geopotential Height, scaling differences.
>
> * **Resolution & Levels:** Differences in spatial resolution and vertical levels.
>
> * **Evaluation Protocol:** Differences in evaluation masks and temporal sampling.
>
> All these factors differ across setups. Without matching these conditions, raw RMSE values are not directly comparable. Again, we do not claim to outperform Pangu-Weather in its 0.25° operational regime. We focus on a different regime—standard ERA5 ML benchmarks—where our method is both **competitive and extremely efficient**.
>
> ***
> ### 2. On citations in Table 2
>
> We sincerely thank you for this suggestion. In the revised version, we will ensure that these references are clearly visible in the table and in the text to properly credit prior work.

---

### Meta-Review · Area_Chair_dD9L · 2026-01-07

**Summary:**

This paper presents a deep learning framework for weather forecasting. The title of the paper is quite misleading in two aspects: the term "climate" indicates long-term weather (not 24h studied in the paper) and the term "LLM" (as pointed out by reviewers) does not make sense. In addition, the reviewers pointed out, the paper studied a quite toy setting with only a 5.625-degree or a 1.5-degree setting (recent weather forecasting models worked on a 0.25-deg setting) and only within a 24h lead time (at least 3d or 5d lead times shall be tested). The AC and reviewers also found that the paper overclaims a lot (e.g., it exceeds state-of-the-art accuracy). Given the current status, it is difficult to judge the value of this work.

**Reviewer Concerns:**

The authors partly addressed the concerns, e.g., raising the resolution from 5.625-deg to 1.5-deg in the rebuttal, but most major concerns remained unsolved.

**Reviewer Scores:**

I think the reviewers will converge on a consensus of rejection. The AC agrees with the negative reviewers that the paper is not ready for publication.

---

### Decision · Program_Chairs · 2026-01-26

Reject